# Outbreaks of H5N1 High Pathogenicity Avian Influenza in South Africa in 2023 Were Caused by Two Distinct Sub-Genotypes of Clade 2.3.4.4b Viruses

**DOI:** 10.3390/v16060896

**Published:** 2024-05-31

**Authors:** Celia Abolnik, Laura Christl Roberts, Christine Strydom, Albert Snyman, David Gordon Roberts

**Affiliations:** 1Department of Production Animal Studies, Faculty of Veterinary Science, University of Pretoria, Pretoria 0110, South Africa; laura.roberts@westerncape.gov.za; 2Department of Agriculture, Western Cape Government, Elsenburg 7607, South Africa; 3Centre for Veterinary Wildlife Research, Faculty of Veterinary Science, University of Pretoria, Pretoria 0110, South Africa; 4SMT Veterinary Laboratory (Pty) Ltd., Irene, Pretoria 0178, South Africa; ts@smtvet.co.za; 5Southern African Foundation for the Conservation of Coastal Birds (SANCCOB), Cape Town 7441, South Africa; albert@sanccob.co.za (A.S.); david.gr@sanccob.co.za (D.G.R.)

**Keywords:** high-pathogenicity avian influenza, clade 2.3.4.4b, H5N1, seabirds, poultry, phylogenetic analysis, PA-X isoform, PB1-F2 truncation, wild bird disease surveillance

## Abstract

In 2023, South Africa continued to experience sporadic cases of clade 2.3.4.4b H5N1 high-pathogenicity avian influenza (HPAI) in coastal seabirds and poultry. Active environmental surveillance determined that H5Nx, H7Nx, H9Nx, H11Nx, H6N2, and H12N2, amongst other unidentified subtypes, circulated in wild birds and ostriches in 2023, but that H5Nx was predominant. Genome sequencing and phylogenetic analysis of confirmed H5N1 HPAI cases determined that only two of the fifteen sub-genotypes that circulated in South Africa in 2021–2022 still persisted in 2023. Sub-genotype SA13 remained restricted to coastal seabirds, with accelerated mutations observed in the neuraminidase protein. SA15 caused the chicken outbreaks, but outbreaks in the Paardeberg and George areas, in the Western Cape province, and the Camperdown region of the KwaZulu-Natal province were unrelated to each other, implicating wild birds as the source. All SA15 viruses contained a truncation in the PB1-F2 gene, but in the Western Cape SA15 chicken viruses, PA-X was putatively expressed as a novel isoform with eight additional amino acids. South African clade 2.3.4.4b H5N1 viruses had comparatively fewer markers of virulence and pathogenicity compared to European strains, a possible reason why no spillover to mammals has occurred here yet.

## 1. Introduction

The goose/Guangdong (Gs/GD) lineage of H5N1 subtype high-pathogenicity avian influenza (HPAI) viruses emerged in China in 1996, and by 2005, it had entered wild migratory bird populations and began to spread intercontinentally via the flyways. Over the next decade, the Gs/GD H5 lineage diversified into numerous clades, which are defined by the phylogeny of the haemagglutinin (HA) gene, and underwent genomic reassortments with other low-pathogenicity avian influenza viruses [1,2]. Gs/GD H5Nx HPAIVs gained the extraordinary capability to be maintained sub-clinically in wild water bird populations. These birds contaminate their environments with enormous quantities of viruses shed in oral fluids and feces that, upon contact, can cause lethal infections in poultry and a wide range of other bird species [1,3].

Clade 2.3.4.4 H5Nx HPAI viruses have become the dominant clade in wild birds since 2014, especially the clade 2.3.4.4 group b viruses that caused the two most recent intercontinental epizootics, in 2016–2017 and 2020–2024 [1,4]. In duck species, clade 2.3.4.4b viruses are shed in significantly increased levels overall and from the cloaca, resulting in more efficient transmission in wild bird populations [5]. Epizootics of clade 2.3.4.4b are also intensifying in magnitude and distribution. The 2016–2017 wave, dominated by an H5N8 subtype, reached the southern tip of the African continent for the first time [6], with 49 countries in Asia, Europe, and Africa reporting outbreaks. The current wave, dominated by H5N1, reached not only southern Africa but also new territories on the South American continent and as far south as Antarctica [4,7,8,9]. To date, 81 countries on all continents except Oceania have reported outbreaks. More alarmingly, in 2023, clade 2.3.4.4b viruses began to increasingly infect mammals in Europe and the Americas, raising fears that a pandemic human strain may soon emerge [10]. Aside from the catastrophic effects of clade 2.3.4.4b H5Nx HPAI on global poultry production, unprecedented outbreaks in wild birds and marine mammals have had dire ecological and conservation impacts [10,11,12,13].

The source of South Africa’s clade 2.3.4.4b H5N1 viruses was ultimately Europe, via West Africa. Palearctic-breeding ducks most likely carried the virus southward to overwintering grounds in the Senegalese region in December 2020. Intra-African movements of wild waterfowl are believed to have then moved the virus further south in their seasonal, rainfall-driven movements [7]. South Africa, Botswana, Lesotho, and Namibia were affected by outbreaks in wild and domestic birds in 2021 [7,14,15,16]. A molecular epidemiological study of the virus genomes from the South African epizootic in 2021 revealed that seven distinct H5N1 HPAI sub-genotypes, namely SA1-SA5, SA7, and SA8, were associated with outbreaks in wild and domestic birds in the first six weeks of the outbreaks. Virus diversity decreased after the winter season, outbreaks in commercial poultry ceased, and only two sub-genotypes, SA13 and SA15, were still present in the country by summer of 2021/2022 [7]. SA13 was exclusively associated with coastal seabird outbreaks along the southern Cape coastline but also spread up the west coast to Namibia to a Cape Cormorant (*Phalacrocorax capensis*) colony on Bird Island, Walvis Bay, in December 2021 [7,15]. SA15 is a unique southern African sub-genotype, initially implicated in outbreaks that started in early June 2021 in Botswana [7,14]. It reached South Africa much later in 2022, where it caused a single outbreak with a high mortality rate in commercial ostriches near the town of Fauresmith in the Free State province in November.

There was a lull in reported H5N1 HPAIV cases in South Africa over the summer of 2022/2023 until May 2023, when H5N1-confirmed cases in coastal seabirds in the Western Cape province started to increase again. The species most affected were the Swift tern (*Thalasseus bergii*) and African penguin (*Spheniscus demersus*), but the virus was also detected in Common terns (*Sterna hirundo*), Kelp gulls (*Larus dominicanus*), Hartlaub’s gulls (*Chroicocephalus hartlaubii*), and an African oystercatcher (*Haematopus moquini*) from along the western and south-western coastline between Lambert’s Bay and Nature’s Valley, near George. The virus was also detected in Swift terns in KwaZulu Natal (KZN) province for the first time in July (wahis.woah.org (accessed on 8 May 2024)). Starting in mid-April 2023, five commercial layer chicken farms in the Paardeberg area, northwest of Paarl in the Western Cape province, became infected with H5N1 HPAI viruses. Only two farms within the small 6 km diameter area of the outbreaks were unaffected. H5N1 HPAI virus infection was next detected on two layer farms near George, Western Cape province, in late May/early June, and from the beginning of July to September 2023, four outbreaks were confirmed in layer breeders in the Camperdown region of KZN, approximately 1300 km from the earlier cluster of cases. A total of 1.75 million chickens were culled in 2023 in the Western Cape and KZN provinces to control the spread of HPAI. From September to November, another six smaller outbreaks, from four more northern provinces, were reported to the World Organization for Animal Health, but the virus was not available for sequencing. At the time, there was speculation that the virus responsible for the coastal seabird mortalities since March had spilled over to the commercial chickens in the Western Cape province, spread from Paardeberg to the George region, and then to KZN with live bird movements. Farmers expressed concerns that coastal seabirds were an HPAI virus reservoir that posed a risk to commercial chicken farms in coastal areas. In the present study, we sequenced and analyzed the genomes of all the available H5N1 HPAI virus outbreak strains from the Western Cape and KwaZulu-Natal provinces in 2023 in order to elucidate the origins and spread of the outbreaks. Concurrently, we conducted a national active surveillance program for avian influenza in wild ducks.

## 2. Materials and Methods

### 2.1. Collection and Testing of Wild Duck Fecal Samples

Wild duck environmental fecal samples were collected and tested during a national active surveillance program in 2023. The procedures for the collection, pooling of 5 fecal swabs, transportation, and testing were as previously described [7]. Sampling in the Gauteng province at the Bon Accord Dam and Johannesburg Zoo Lake sites in February and March 2023 was undertaken by the University of Pretoria (UP) and other samples were collected by poultry veterinarians across the country, targeting dams and wetlands in the vicinity of poultry farms. If any sick or dead wild birds were found, veterinarians collected tracheal and cloacal swabs or tissues. No sampling was conducted in January or December 2023 due to the summer holiday period. Total nucleic acids were extracted from fecal swab fluids using IndiMag Pathogen kits in an IndiMag^TM^ 48 instrument (Indical BioSciences, Leipzig, Germany) and screened for the presence of viral matrix (M) and nucleocapsid protein (NP)-specific RNA using real-time reverse transcription PCR (rRT-PCR) with VetMAX^TM^-Gold AIV Detection Kits (ThermoFisher Scientific, Waltham, MA, USA), according to the kit instructions in a StepOnePlus instrument (ThermoFisher Scientific). Samples with cycle threshold (Ct) values < 40 were considered as positive. For subtyping of positive cases, VetMAX Plus RT-PCR kits (ThermoFisher Scientific) were used with subtype-specific primers and probes for H5 [17], H7 [18], H9, H11 hemagglutinin (HA), and N6 (neuraminidase) subtypes [19]. For H6, the forward, reverse, and probe sequences in 5′ to 3′ orientation were as follows: H6 CkSA FOR: GAT GCA AAT GTG AAG AAC CTA TA; H6 CkSA REV: TTT GAC AGA CTC CAT GCA ATC; H6 CkSA PRO: FAM-GCT TTG ART TCT GGC ATA AAT GTG-MGB. These H6 oligonucleotides specifically detect the endemic South African H6N2 lineage viruses and may have reduced sensitivity for other Eurasian type H6 viruses [20]. The rRT-PCR thermal cycling conditions were used for all subtyping assays were as follows: 48 °C for 15 min; 95 °C for 10 min; 40 cycles of 95 °C for 15 s and 50 °C for 45 s.

### 2.2. Poultry Samples

H5N1 HPAI outbreaks in commercial chickens and any avian influenza virus (AIV) in commercial ostriches were diagnosed by accredited veterinary laboratories using validated rRT-PCR-based methods [21]. RNA extracts from the swabs or tissues of confirmed H5 clinical cases in chickens or AIV-positive cases in ostriches were forwarded to UP for genome sequencing. Our aim was to include at least one representative virus from each of the H5N1 outbreaks, where possible. AssureCloud (Pty) Ltd. Laboratories, Cape Town, South Africa, in Oudtshoorn and Centurion and SMT Veterinary Laboratory (Pty) Ltd. in Midrand, South Africa contributed RNA to this study on the authorization of their poultry industry clients.

### 2.3. Coastal Seabird Samples

Data on suspected HPAI cases in seabirds, which were not laboratory-confirmed, were obtained from the Department of Forestry, Fisheries and Environment online reporting tool, designed and managed through the National Oceans and Coasts Information Management System. Suspected and confirmed HPAI cases in seabirds were reported, mostly by staff of the SANCCOB seabird rehabilitation center in Table View, Cape Town, and City of Cape Town municipal staff, from January to September 2023 (n = 151). Between one and three cases were reported monthly from January to April and from August to September, but there was an increase in reported cases from May to July, with a peak of 107 cases in June. Most of the June cases (n = 87) were Swift terns, and most were from Simon’s Town, but reports were widely distributed in May and June. Most affected Swift terns, where age was reported, were juveniles (51/66). African penguin cases peaked in June and July, with 9/15 cases in Simon’s Town, though three other colonies were represented. Samples were taken at SANCCOB and at the African Penguin and Seabird Sanctuary (APSS) in Gansbaai. Seabirds were sampled either because they showed clinical signs of HPAI (usually neurological signs such as head twitches or seizures) and/or did not demonstrate a visible cause of death on postmortem examination. SANCCOB submitted brain samples in viral transport medium (as described in [7]) for testing, and the samples submitted for two penguins from Dyer Island, by APSS, comprised both a tracheal and cloacal swab pool and an organ pool. AssureCloud Laboratory’s Oudtshoorn branch performed the official AIV diagnosis on the coastal seabird samples. Samples were screened using the rT-PCR RealTime ready RNA Virus Master kit (Roche Life Science, Basel, Switzerland), targeting the IAV matrix gene with primers and probes [22], supplied by TIB Molbiol Synthesis Laboratory (Berlin, Germany). Ct values < 35 were considered positive. AIV-positive samples were tested with H5-specific primers and probes [18]. Ct values < 35 were considered positive. RNA was forwarded to UP for genome sequencing.

### 2.4. Genome Sequencing and Bioinformatic Analysis

Genome amplification RT-PCR was performed on all samples with Ct values < 30, as previously described [17,23]. The RT-PCR amplification products underwent Ion Torrent next generation sequencing (NGS) at the Central Analytical Facility of Stellenbosch University. Raw data were analyzed at UP, where the methods used for genome assembly, sequence analysis, distance matrices, concatenation, maximum likelihood phylogenetic trees, and reconstruction of Maximum Clade Credibility (MCC) trees were all performed as previously described [7]. The exception was for the concatenated genomes, where the output trees of the chain length of 70 million required to achieve an Estimated Sample Size of >200 were down-sampled in the BEAST 2 package’s LogCombiner v2.6.7. program prior to tree annotation. Statistical analysis of Ct values was performed in GraphPad Prism version 10.2.1 software. The genomic sequences produced in this study were deposited in the GISAID EpiFlu database (https://gisaid.org/ (accessed on 8 May 2024)).

## 3. Results

### 3.1. Molecular Epidemiology of H5N1 HPAIVs in South Africa in 2023 in Poultry and Wild Birds

The genomes of H5N1 influenza A viruses, from 21 samples from nine commercial chickens outbreaks and 37 samples from wild birds in South Africa in 2023, were phylogenetically analyzed. The wild bird samples represented 95% (37/39) of AIV PCR-positive samples from sick or dead birds in the Western Cape province, and 36/38 seabird viruses that were sampled during the peak in suspected cases, from May to July. The concatenated genomes were analyzed in a time-scaled maximum clade credibility (MCC) tree (Figure 1 and Appendix A) and the individual genome segments were analyzed in Maximum Likelihood phylogenetic trees (Appendix A). All wild bird viruses originated from coastal seabirds except for a single virus from an Egyptian goose (*Alopochen aegyptiaca*), with neurological clinical signs, that was sampled in the Western Cape province (A/Egyptian goose/South Africa/249/2023). These phylogenetic data show that the 2023 outbreak viruses were clearly split between two groups, sub-genotypes SA13 and SA15, that were previously identified in the 2021–2022 outbreaks [7]. The SA13 sub-clade still comprised coastal seabird viruses exclusively, whereas all the poultry viruses and three gull viruses fell within sub-genotype SA15. No reassortment events with other LPAIVs were detected in any of the viruses we analyzed (Appendix A). The two sub-clades will be described separately below.

#### 3.1.1. Sub-Genotype SA13

The cluster of coastal seabird cases in 2023 were not descended from the latest viruses sequenced in 2022, i.e., those from outbreaks in African penguins in October and November. Instead, the most recent common ancestor was A/Common tern/South Africa/22060305/2022, sampled much earlier in the Cape Town area, in June 2022. The long branch connecting the 2023 cluster with the aforementioned ancestral virus indicates a prolonged period of undetected circulation of this specific strain. The relatively low genetic variation within the 2023 cluster points to a clonal expansion of an ancestral virus, which, according to the tMRCA analysis, emerged in early March 2023 (95% highest posterior density (HPD) February to March 2023) (Appendix A). The earliest viruses in the 2023 outbreaks, namely A/Kelp gull/South Africa/K137/2023 and A/Common tern/South Africa/CMT004/2023, were sampled at the end of March and formed an outgroup to the rest of the viruses from 2023. Three genetic sub-clades, designated as A, B1, B2, and C (Figure 2), could be identified in the 2023 coastal seabird viruses, with all groups circulating concurrently.

Group A comprised African penguin viruses (n = 5), a Swift tern virus, an African oystercatcher virus, and a Common tern virus. Group A viruses were widely distributed (Figure 3), and 6/8 were detected during the peak mortality in June. Group B1 consisted of African penguin viruses (n = 3), Common tern viruses (n = 2), and a Swift tern virus, and group B2 consisted of two African penguin viruses and a Common tern virus. Group B viruses were also widely distributed, and 6/9 were detected in June. Group C comprised Swift tern viruses (n = 9), African penguin viruses (n = 3), a Kelp gull virus, and a Common tern virus and were only detected closer to Cape Town, with 6/14 sampled from Simon’s Town (Figure 3). Seven of fourteen were detected in May and six in June.

#### 3.1.2. Sub-Genotype SA15

The SA15 viruses from 2023 formed three distinct sub-clades, designated here as groups D, E, and F (Figure 4). Group D contains the first 2023 case of the SA15 sub-genotype, A/Egyptian goose/South Africa/249/2023. The viral sequence was obtained from the tissues of an Egyptian goose at the Vrolijkheid Nature Reserve near McGregor, Western Cape province, on 18 April, just two days before the first case in the layer hens was reported in the Paardeberg region, approximately 100 km away. It was one of approximately 75 Egyptian geese observed at the site, but no other suspected cases were reported. The viruses from the poultry outbreaks were closely related to the Egyptian goose virus, with an RCA that is dated very shortly before, in March 2023 (95% HPD February–April) (Figure 4 and Appendix A).

Group E consists of the gull viruses (two Kelp gulls and a Hartlaub’s gull) from north of Cape Town, about 35 km east of the Paardeberg, and chicken viruses that were detected in George, Western Cape province, approximately 350 km east. The two chicken viruses in group E, A/chicken/South Africa/717619/2023 and A/chicken/South Africa/719311/2023, caused outbreaks a month apart in May and June and shared an RCA. Since only a partial genome was recovered for one of the two outbreaks in George, a Maximum Likelihood phylogenetic tree derived from six concatenated genome segments is also included in the supplemental data (Appendix A).

Relatively longer branches within group E point to higher genetic variability, and the inclusion of three gull viruses points to wild birds being likely the source of the chicken outbreaks. The phylogenetic data furthermore shows that these outbreaks in George were not directly related to those in wild birds and poultry in the McGregor/Paardeberg regions. Groups D and E, however, share an RCA dated at early March 2023 (95% HPD March–April), suggesting that wild hosts carrying these the viruses arrived in the Paardeberg, McGregor, and George regions of the Western Cape province almost simultaneously in March 2023.

The phylogenetic data unequivocally determined that the outbreaks in KZN commercial chickens from July to August 2023 (group F) were genetically unrelated to the Western Cape cases (groups D and E). Closely related strains EP232013 and EP232014 were from 27-week- and 53-week-old layers, respectively, at different sites of the same producer; therefore, secondary spread within the company is a strong possibility in this case. The larger genetic distances between the aforementioned outbreaks that occurred in late August 2023 and two others that occurred in early and late July, namely A/chicken/South Africa/720743/2023 and 722490/2023, from a different producer, were unrelated, however. The RCA for the Western Cape outbreaks (groups D and E) and the KZN outbreaks (group F) is dated at mid-September 2022 (95% HPD February to June 2022–January 2023) (Appendix A). This was two months before the index case of SA15 in South Africa in November 2022, on the Free State province commercial ostrich farm [7]. In turn, the RCA for all South African SA15-related outbreaks in 2022–2023 is late July 2022 (95% HPD April–October 2022).

### 3.2. Environmental Wild Bird Surveillance

A total of 477 wild duck environmental fecal swab pools, representing ±2385 individual swabs, were received for testing from six of the nine South African provinces. No samples were received from the Free State, Limpopo, or Northern Cape provinces (Figure 5; Table 1). The majority of samples originated from the Gauteng, Western Cape, and KZN provinces, which have the highest poultry densities. Overall, 287 pools tested AIV positive (60.1%), and Gauteng had the highest prevalence (80.9%), followed by KZN (55.7%) and the Western Cape (44.8%). A breakdown of cases per month per province is provided in the supplemental data (Appendix A). It must be noted that the sampling period in Gauteng was purposefully targeted to the late summer/autumn months (February–March), when AIV prevalence peaks in ducks in this summer rainfall region [17].

Of the total AIV positive samples, 42.2% were identified as the H5 subtype, and among the three provinces with the highest number of samples tested, Gauteng had the highest H5 subtype proportion (57.1%), followed by the Western Cape (32.9%) and KZN (20.5%).

The highest incidence of H5 in Gauteng was detected in February (47.4%) and March (69.8%), but H5 was also detected in Gauteng in April, May, and August (Appendix A), and no samples were tested after August. In the Western Cape, H5 was detected in April (62.5%), June (36.4%), and July (57.1%), corresponding with elevated precipitation in this winter rainfall region, but not in the period before or after this. Unfortunately, no samples from March were tested, when the SA15 virus is thought to have been introduced to the region. In KZN, also a summer rainfall region, H5 was detected in April, June, August, and September, in incidences ranging from 20 to 66.6% of AIV positive cases. Only two (0.7%) H7-positive pools were identified from wild bird samples in South Africa in 2023, one from the Camperdown region of the KZN province in April and the other from the Oudtshoorn region of Western Cape province in October. Neither of these tested positive for the N6 subtype (an H7N6 HPAI outbreak occurred in South African poultry in 2023; wahis.woah.org (accessed on 8 May 2024)). No chicken-lineage H6 subtype was detected in the South African wild duck samples in 2023, but one H11 subtype virus was identified at Muldersdrift in the Gauteng province (sampling date 25 April 2023), and one H9 virus was identified in Oudtshoorn, Western Cape province (sampling date 3 October 2023). The real-time PCR data also enabled a comparison of the relative sensitivity of the AIV and H5 subtype detection assays. The mean Ct value for the AIV group detection assay was 34.3 ± 2.29 for samples later classified as H5-positive, whereas the value for the H5 assay was more than a log magnitude higher at 35.7 ± 2.47 (Figure 5). Since only two samples were H7-positive, they were excluded.

### 3.3. Identification of AIVs in Ostriches

Two AIV-positive tracheal swab pools, collected from commercial ostriches in the Western Cape province, had sufficiently low Ct values (<30) for AIV genome amplification and sequencing. The closest relatives to all eight genome segments were identified by BLAST analysis (Appendix A). A/ostrich/South Africa/AI9145-P42/2023, sampled in July 2023 on a farm near Oudtshoorn, was identified as a Eurasian-lineage H6N2 strain. A/ostrich/South Africa/761940/2023, sampled in May on a farm in the Ladismith area, was identified as an H12N2 strain. In both viruses, some of the genes shared RCAs with viruses previously identified in wild birds or ostriches in the southern African region. Ostriches have frequent contact with wild birds in their camps and are highly susceptible to infection with AIVs. They act as sentinels for viruses that are present in the wild bird population. H6N2 and H12N2 viruses were evidently circulating in wild birds of the Western Cape province in 2023.

### 3.4. Molecular Markers of Host Tropism, Increased Virulence and Resistance to Antivirals

The translated protein sequences of all South African H5N1 HPAI viruses from 2021 to 2023 outbreaks were examined for known molecular markers (amino acid mutations) of pathogenicity and mammalian adaptation. The HA proteins contained only two mutations which are associated with increased binding of the virus to mammalian-type α2-6 sialic acids [24], namely T156A and V182N (H5 numbering), but these mutations are also common in the clade 2.3.4.4 viruses circulating in other continents [4,25]. The H5N1 viruses that circulated in Europe in 2021–2023 contained comparatively more mutations in HA that confer the ability to bind α2-6 sialic acids, such as S123P paired with R497K, S133A, S154N, S155N, K218Q paired with S223R and E251K, S107R paired with T108I, and K394E, that confer other virulence traits [24,25]. Polymorphisms in the NA protein that confer antiviral resistance [24] were not detected in any South African virus sequences.

The PB2 proteins contained multiple mutations that are associated with enhanced polymerase activity and consequently increased virulence in mammals. These include a pairing of L89V, G309D, T339K, R447G, I495V, K627E, A676T, K251R, G309D, T339K, Q368R, K389R, H447Q, R477G, I495V, V598T, K627E, and K676T singly. K627E is specifically associated with increased virulence in chickens [24,25,26]. However, the aforementioned mutations are also described in clade 2.3.4.4 viruses from other regions and are not unique to the South African strains [24,26]. A number of virulence-associated mutations that are present in European viruses were absent from the South African strains, namely E249G, I292V, D701N, and T631I [24,25,26].

Of the fifteen mutations or mutational combinations in the PB1 protein associated thus far with increased polymerase activity and/or viral replication in mammals [24], only A/ostrich/South Africa/21070586/2021 (H5N1) contained a K577E mutation, and all contained D3V and D622 mutations, but the latter two mutations were also present in other European clade 2.3.4.4 viruses [25].

The N66S mutation in the PB1-F2 sequence, which is associated with enhanced replication, virulence, and antiviral responses, and the T51M mutation, which is associated with decreased polymerase activity, replication, and virulence in ducks [24], were present in European clade 2.3.4.4 H5N1 viruses [25] but were absent from all South African strains. Most interestingly, the SA15 viruses, with one exception (A/chicken/South Africa/722490/2023; group F), had a truncated PB1-F2 protein, which is associated with increased polymerase activity and virulence in mice [27]. All other South African H5N1 viruses expressed a full-length PB1-F2 protein.

Of twenty-one mutations or mutational combinations in the PA protein known to confer increased virulence in mice or mammalian cells [24], eight of which are present in European clade 2.3.4.4 H5N1 viruses [25], only four were present in all South African strains, namely S37A, P190S, N383D, and N409S. Some of the viruses in sub-genotypes SA10, 11, and 12, which circulated in chickens in 2021–2022 [7], additionally contained the K142E mutation, which is associated with increased virulence in mice, but this mutation was not present in any viruses from 2023 that were sequenced in the present study.

In the PA-X protein, the majority of South African strains since 2021 contained the untruncated, full-length 252 amino acid sequence. An exception was noted in the SA15 sub-genotype, specifically the viruses that caused the Western Cape outbreaks in 2023 (groups D and E, Figure 4), but not group F viruses. In these viruses, a transitional mutation of two adjacent adenine nucleotides at position 769–770 (PA gene numbering) to guanine converted the stop codon and the subsequent thymine (T) amino acid to tryptophan (W) and alanine (A), respectively. These PA-X proteins are consequently extended at the C-terminal by eight amino acids, with the sequence WAPELSHF*.

The NP genes of all South African H5N1 viruses, including those in 2023, contained two mutations that are demonstrated to increase virulence in chickens, namely M105V and A184K, but none of the other mutations known to increase polymerase activity in mammalian cells [24]. The K198R mutation, associated with decreased polymerase activity in mammalian cells [24], was present in the coastal seabird viruses of SA13 and a single virus from 2021, A/cormorant/South Africa/21120147/2021 (H5N1), but was absent from all other South African H5N1 viruses since the outbreaks started in 2021. A single virus from 2022, A/Cape gannet/South Africa/702625 G123/2022 (H5N1), contained the N319K mutation associated with increased polymerase activity, whereas it was absent from all other SA viruses in 2021–2023. One NP mutation, I41V, found in European clade 2.3.4.4b viruses, was absent from all South African strains. This mutation is also responsible for increased polymerase activity and, hence, virulence [24,25].

Three of the four M1 protein mutations that promote virulence in mice, which were present in European H5N1 viruses since 2021 [25], are also present in the South African viruses, namely N30D, I43M, and T215A. T139A, which is present in the clade 2.3.4.4 viruses of other regions, was, however, absent from South African viruses. None of the mutations in M2 that confer a reduced susceptibility to amantadine and rimantadine [24] were present in the South African strains, with the exception of eight viruses sequenced in the previous study [7], namely A/Cape cormorant/South Africa/21110001/2021, A/chicken/South Africa/JB2201/2022, A/chicken/South Africa/DW2201/2022, A/chicken/South Africa/DW2202/2022, A/European white stork/South Africa/BA107/2022, A/chicken/South Africa/PRL118/2022, A/chicken/South Africa/697683/2022, and A/Cape cormorant/South Africa/21110015 D2095/2021. In the aforementioned viruses, S31N, a mutation that increases resistance to amantadine and rimantadine, was present.

The NS1 protein is a notorious virulence antagonist, containing at least 16 mutations or mutational combinations associated with increased virulence or decreased antiviral activity [24]. The presence of P42S, L103F paired with I106M or I106M alone, and V149 was observed in all clade 2.3.4.4 South African as well as European strains since 2021, and these mutations are associated with an increased virulence and decreased antiviral responses in mice [24,25]. Only six viruses from a variety of wild and domestic species in the local 2021 outbreaks contained the D74N mutation that enhances viral replication in mammalian cells [24], and only one virus, A/African fish eagle/South Africa/21060065/2021 (H5N1), contained a D92E mutation that increases the virulence in swine and mice [24]. The majority of South African viruses since 2021 also contained the C138F mutation, which increases replication in mammalian cells and downregulates the interferon response [24]. The notable exception was the KZN outbreaks strains from 2023 (group F), which contained a C138L mutation, the phenotypic effect of which is unknown. Like many other clade 2.3.4.4b viruses, all South African viruses contained the C-terminal sequence of 227-ESEV-230, which, in other H5N1 viruses, decreased viral replication in mammalian and avian cell lines, but in H1N1 and H7N1 viruses, increased viral replication and virulence in mice [24]. Finally, all SA viruses contained a T47A mutation the NS2/NEP protein, which, when paired with a N205S mutation in NS1, which is present, was previously demonstrated to decrease antiviral responses in ferrets [24].

### 3.5. Unique Markers or Unknown Phenotype That Emerged in the Coastal Seabird Lineage SA13

The SA13 sub-genotype that first emerged in 2021 and persisted to 2023 only affected coastal seabirds. We therefore identified and tabulated the unique amino acid mutations that may have played a role in this host species adaptation (Appendix A). Some unique mutations that emerged in 2021 were retained to 2023, for example PB2-E6K, HA-A102V, and others, whereas some other mutations first emerged in 2022, for example NP-M371I and NS1 V84M. A third category of mutations was unique to the SA13 viruses in 2023, for example PB2-T238N. In contrast to 16 unique mutations distributed across the SA13 viruses’ genomes in 2021, by 2023, 40 mutations had emerged, 92.5% of which were found in all strains. None of these unique markers is known to be associated with any specific phenotypic traits, apart from NP-K198R, which was already present since 2021 and is associated with decreased polymerase activity in mammalian cells [24]. HA-E201K occurs near the receptor binding site in the 190 helix (H3 numbering) and may influence receptor binding in the late SA13 viruses. The proteins that accumulated the most changes over time, proportionally, were PA, with six mutations in 2023 vs. two in 2021, and NA, with six mutations, none of which were present in 2021. The NA- I267V and V346I mutations first emerged 2022 but were only present in some of the viruses.

The HA and NA proteins play crucial roles in host cell receptor binding and infection and in virus release, respectively. A102V in the HA protein emerged early in SA13 viruses and was retained. E201K was present in some viruses in 2021 and all viruses in 2022, but only in the group B1 viruses and A/Common tern/South Africa/CMT004/2023. S336N only emerged in 2023 but was present in all viruses. All three mutations in HA occurred in the globular head region where most antigenic epitopes and the receptor binding site are located. In the NA protein, one of the unique mutations that emerged in 2023, I7T, occurred in the NA transmembrane region, I40R and N42I occurred in the stalk region, and three new mutations occurred in the globular head region, namely I267V, V346I, and S450G.

When the SA13 viruses from 2022 were compared to the SA13 viruses from 2023 using amino acid sequence distance matrices of the of H5 HA protein (576 aa; Appendix A) and N1 NA (470 aa; Appendix A), the mean number of mutations in the HA protein was 2.1 (1–5), but in the NA protein, it was 8.1 (1–13). Percentagewise, the mean amino acid mutations in HA in SA13 increased by 0.36% between 2022 and 2023, and in NA, the value was 1.72%. Overall, there were 4.7-fold more mutations in the NA protein compared to the HA protein. For comparison, when the SA15 viruses from 2023 were compared with all other sub-genotypes from 2021–2022 (SA1 to SA14), the mean number of mutations in HA was 6.4 (3–11), and in NA, it was 20.5 (18–25), which is anticipated since SA15 is a phylogenetically distinct virus, but there were only 3.2-fold more mutations in NA compared to HA in these vastly different sub-genotypes.

## 4. Discussion

Detections of clade 2.3.4.4b H5N1 HPAI viruses, identified in South Africa since April 2021, waned again by the summer in late 2022 but resumed in March 2023 in the Western Cape province, with a peak in June. Confirmed cases in coastal seabirds from around Cape Town came first, followed a month later by a case in an Egyptian goose inland and a localized outbreak in commercial layer hens. When a new cluster of poultry outbreaks appeared in George in May, followed by another in the KZN province in July, secondary spread was suspected. To investigate the source/s of these outbreaks, we sequenced the viral genomes to trace the epidemiology of these new H5N1 outbreak viruses. The sequence analysis clearly distinguished the 2023 epizootic in coastal birds from those in poultry in South Africa. Apart from three gull viruses found just north of Cape Town, within 40km of the affected Paardeberg poultry farms, all coastal seabird viruses were identified as a resurgence of sub-genotype SA13. In contrast, all the outbreaks in chickens were caused by sub-genotype SA15. SA15 is a unique southern African variant that was first associated with outbreaks in Botswana in mid-2022 [13] and was the cause of a single outbreak in commercial ostriches in South Africa’s Free State province in November 2022 [7]. Furthermore, the three clusters of SA15-associated poultry outbreaks in the Paardeberg and George areas and KZN were phylogenetically unrelated.

The similar timing of the outbreaks in the Western Cape seabirds and the chickens may have been due to the heavy rainfall that occurred in the southwestern part of the province in early March and again throughout June. Cape Town International Airport had received rainfall equal to the annual long-term average (500 mm) by mid-June, and the amount usually received by the end of August, after the winter rainfall period (approx. 375 mm), had already been reached by mid-May (E Heyneke, South African Weather Services, pers. comm.). High rainfall in March may have attracted wild waterfowl that introduced SA15 virus to poultry in the Paardeberg area and possibly also to the gulls. Given the narrow temporal range of the poultry farm outbreaks and the close genetic relationship between the viruses, it is not possible to determine if the infection spread between Paardeberg farms or if there were separate introductions by wild birds. Four of the five affected poultry farms were also infected with the H5N8 virus in 2017, which suggests this area is prone to AIV introductions. The gulls infected with SA15 viruses were found relatively close to the Paardeberg area, but there were no mass mortalities and no further detections, which suggests incidental exposure. Hartlaub’s gulls frequent coastal wetlands up to 30 km from the coast and Kelp gulls can be seen on agricultural land, up to 50 km inland [27], which makes contact with waterfowl and their habitats possible. The peak in seabird mortalities from May to July involved all three SA13 subclades, which had been detected previously, so a new virus introduction may not be the direct cause, but rough weather may have increased susceptibility. Juvenile Swift terns, that would have fledged in April/May and would have still been dependent on their parents [28], may have succumbed to a combination of cold, wet weather, and their first exposure to HPAI virus.

The progenitors to the H5N1 outbreaks in the two affected provinces in 2023 had evidently persisted in the wild bird reservoir over the previous summer, but no large wild bird mortalities, other than those reported in this study, were recorded. The persistence of H5N1 in the wild reservoir is supported by active wild bird surveillance data. Of all AIV positive wild bird fecal samples, 42% tested positive for the H5 subtype, but the actual proportion may be slightly higher due to the difference in sensitivity/specificity of the AIV group detection rRT-PCR vs. the H5 subtyping assay. It was not possible to confirm which of these H5-positive samples were clade 2.3.4.4 H5N1 viruses and which were other Eurasian low-pathogenicity viruses, but H5 viruses were certainly the dominant subtype and were present in most of the provinces in 2023. Surprisingly, Gauteng, where a high incidence of H5 was detected (especially in February and March) has, like the Western Cape and KZN provinces, a high-density poultry producing area, yet only three H5 outbreaks on small-scale farms were reported there in October and November 2023. An H7N6 HPAI outbreak affected Gauteng and surrounding provinces in 2023, but this will be reported on elsewhere. There was no evidence that HPAI or LPAI H7N6 viruses circulated in high numbers in wild birds in 2023; according to our surveillance data. 57% of AIV-positive wild bird samples were subtypes other than H5 or H7, and we identified H9Nx and H11Nx in two cases. H6N2 and H12N2 viruses detected in ostriches indicated that these subtypes were also circulating in the local wild bird reservoir in 2023. The identity of the other subtypes could not be established, but H1Nx, H2Nx, and H10Nx viruses are likely as these were identified in Zambian wild birds in recent years [29].

In an event that was remarkably similar to clade 2.3.4.4b H5N8 in 2017–2018, a sub-genotype of H5N1 viruses that was restricted to seabird species, SA13, emerged in South Africa in 2021 and spread up the coastline to Namibia. The worst affected South African species in 2018 was the Swift tern (approximately 5400; [30]), and in 2021, it was the Cape cormorant (approximately 24,000; [7]). The largest number of suspected cases in 2022 was in African penguins (n = 87), though less than in 2021 (n = 334; Roberts unpublished data), and these were the origin of the last group of viruses detected in 2022 [7]. The total number of suspected cases in seabirds in 2023 was similar to that in 2022, but there was a definite shift from penguins to Swift terns. The eighty-two recorded Swift tern deaths in 2023 were relatively low compared to other years and species, but it was the highest number of Swift tern deaths recorded since 2018. It is therefore interesting that the viruses detected in 2023 were most closely related to a Common tern virus detected in June 2022 and not to more recent penguin viruses. Confirmed and suspected Common tern cases may have been higher than recorded. Approximately 40 old carcasses, mostly juveniles, were counted on the west coast in early June, but date and cause of death could not be determined so they were excluded.

Overall, in 2023, 37 unique amino acid markers spread across all proteins, except for the NS1/NEP, had emerged in the coastal seabird lineage SA13, which distinguished it from all the other clade 2.3.4.4 viruses that circulated in SA in 2021 to 2023. We found that the SA13 neuraminidase gene in particular was prone to antigenic drift. Immune adult birds may have circulated the virus within colonies, and we speculate that the increase in tern mortalities may have been due to the emergence of a novel antigenic variant that was able to overcome the preexisting immunity to SA13 strains that circulated in 2021/2022. It was possibly assisted by harsh weather that increased the vulnerability of juvenile birds.

It is encouraging that no endangered seabird species appeared severely affected by HPAI in the Western Cape in 2023. It is hoped that herd immunity is now protective against outbreaks of the endemic SA13 viruses. However, the recurrence of deaths in Swift terns, although in relatively low numbers, demonstrates the potential for the viruses to either break through waning herd immunity, or mutate to start evading existing immunity. The effect of a future novel, freshly introduced virus is also difficult to predict. Environmental factors and the general health status of the birds are also likely to play a role in their resistance to the virus and underline the need for ongoing general conservation efforts to preserve their environment and food sources.

The molecular dating estimated that SA15 was first introduced to South Africa *circa* early July 2022. The virus evidently persisted in the wild bird populations over the course of a year, spreading southwards to the Free State province by November 2022, and then further south to the Western Cape province by April to June 2023, with simultaneous spread and persistence in the KZN wild bird population by July and August 2023. It is plausible that acquired immunity in wild ducks towards sub-genotypes SA1 to SA12 that circulated in 2021–2022 was overcome by the antigenically different SA15 viruses, leading to elevated replication levels of the virus, increased environmental contamination, and hence, the spillover to poultry in these provinces. There are some indications that SA15 may be more virulent than SA1–SA12, although in vivo pathogenicity testing would be required to verify this. Poultry veterinarians observed that the virus causing the Paardeberg outbreaks caused higher mortalities in layers than outbreaks in the previous year (M. Oosthuizen, pers. comm.), which were caused by sub-genotypes SA4 and 5 [7].

The PA-X protein may have contributed towards the perceived increase in virulence of SA15. The PA-X gene encodes an endoribonuclease that selectively degrades host mRNAs to cause host protein synthesis shutoff, including that of the host’s immune recognition. The PA-X protein is consequently a potent virulence factor that promotes the virus’ own replication [31,32]. The PA-X protein is produced by a ribosomal frameshift during PA translation, fusing the PA N-terminal 191 amino acid (aa) leader sequence with the last 61 or 41aa of the C-terminal, termed the “X-open reading frame” (X-ORF). About 75% of all known influenza A viruses have a 61-codon X-ORF to encode a full-length protein of 252 aa. The remaining 25%, generally limited to the 2009 human pandemic H1N1, triple reassortant swine H1N1, and some other mammalian viruses, bear a 41 amino acid X-ORF to encode a truncated protein of 232 aa [32]. Gao and coworkers [31] conducted truncation experiments on the C-terminal of PA-X using a 2009 H1N1 human pandemic virus, an avian H9N2 virus, and a clade 2.3.4 H5N1 virus, demonstrating in all cases that the last 233–252 amino acids of the PA-X protein increase viral replication and virulence, strengthen the viral-induced inflammatory response and apoptosis, and elevate the host-shutoff ability. All of the South African clade 2.3.4.4b H5N1 viruses from 2021 to 2023 contained the full-length PA-X protein of 252 aa, except for the SA15 Western Cape viruses. The PA-X of the Western Cape SA15 strains all contained an additional eight amino acids that extended the total length to 260 aa. This extension was absent from the Botswana viruses, the Free State ostrich case from 2021, and the 2023 KZN outbreak viruses. This PA-X isoform, designated as PA-X8, was previously also identified in the sequences of least 45 H9N2 subtype viruses in Bangladeshi poultry from 2016–2019, deposited in GenBank [33]. The phenotypic effect of the PA-X8 isoform of the SA15 viruses remains to be experimentally assessed. Another interesting finding that may have a bearing on viral pathogenicity (or attenuation) was that all but one of the KZN SA15 viruses bore a PB1-F2 truncation. PB1-F2 is an accessory protein encoded on the PB1 segment, in the +1 alternate open reading frame. PB1-F2 targets cellular membranes and plays a significant role in viral pathogenesis by modulating the host innate immune response. It induces apoptosis in immune and epithelial cells, down-regulates inflammatory responses, and enhances viral polymerase activity. The effects of PB1-F2 differ between virus strains and host species, as do different lengths of PB1-F2 [34]. The effect of the PB1-F2 truncation on SA15 KZN strains would also make an interesting comparative study to shed light on the role of this protein.

Overall, the South African clade 2.3.4.4b H5N1 viruses detected in 2023 continued to be well-adapted to avian species, retaining a preferential binding for avian-like receptors. In fact, European clade 2.3.4.4b viruses contained proportionally more markers that are associated with host pathogenicity and mammalian adaptation across their genomes. This might also partially explain why, in spite of large numbers of dead coastal seabirds, no sea mammal infections were reported along South Africa’s coastline, unlike other countries where H5N1 viruses have caused mass deaths in marine mammals [4]. Although there appears to be a low zoonotic risk currently, with the clade 2.3.4.4b H5N1 HPAI strains that have reached southern Africa thus far, the situation should be closely monitored.

## Figures and Tables

**Figure 1 viruses-16-00896-f001:**
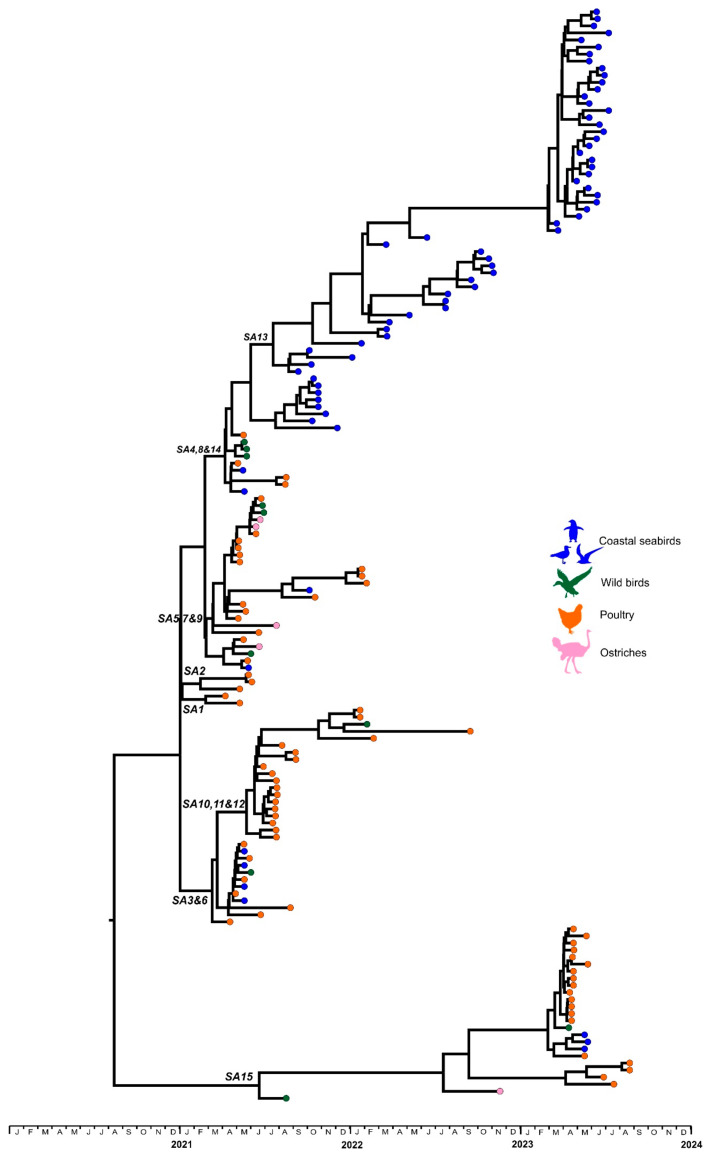
Time-scaled maximum clade credibility tree of the complete genomes of southern African clade 2.3.4.4B H5N1 HPAI viruses. Sub-genotypes of the clades are in italics.

**Figure 2 viruses-16-00896-f002:**
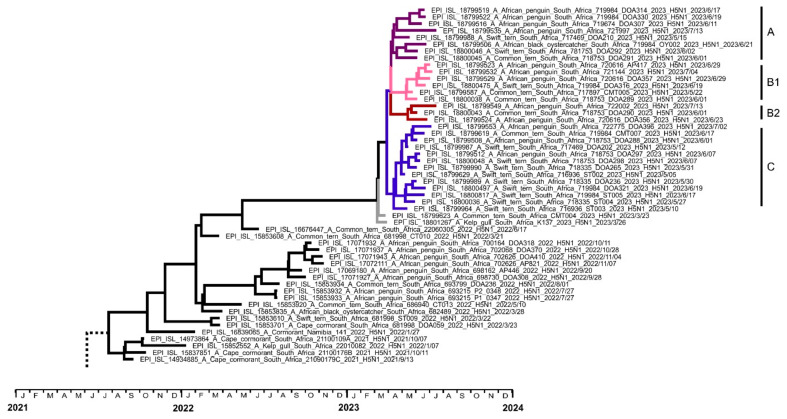
Enlarged view of the concatenated genome time-scaled MCC tree showing sub-genotype SA13 viruses. Sub-clades designated as groups A, B (B1 and B2), and C are indicated.

**Figure 3 viruses-16-00896-f003:**
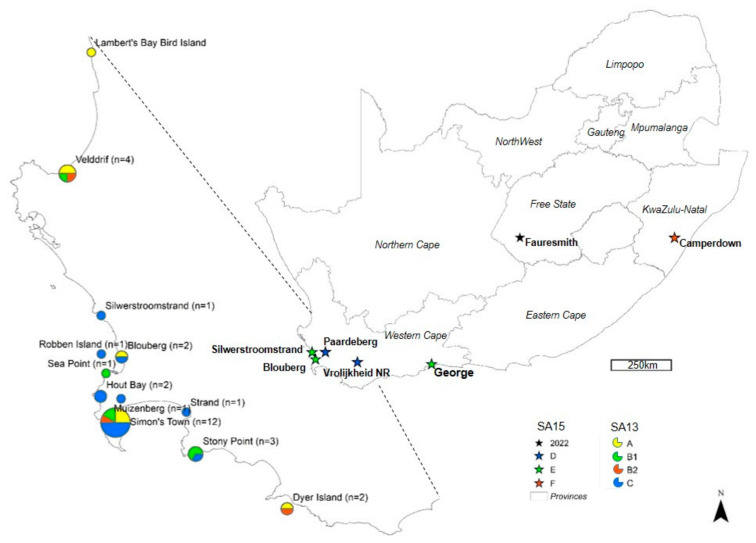
Map showing the locations of clade 2.3.4.4b H5N1 outbreaks in South Africa in 2023.

**Figure 4 viruses-16-00896-f004:**
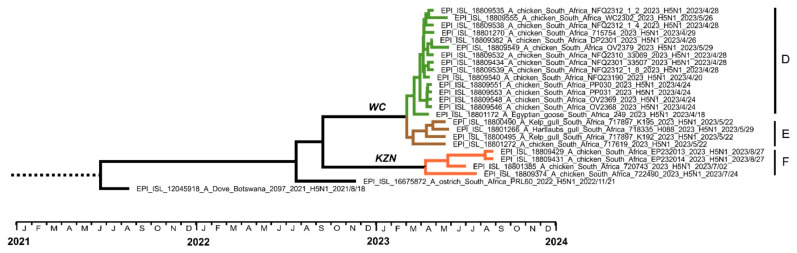
Enlarged view of the concatenated genome time-scaled MCC tree showing sub-genotype SA15 viruses. Sub-clades designated as groups D, E, and F are indicated. WC—Western Cape province; KZN—KwaZulu-Natal province.

**Figure 5 viruses-16-00896-f005:**
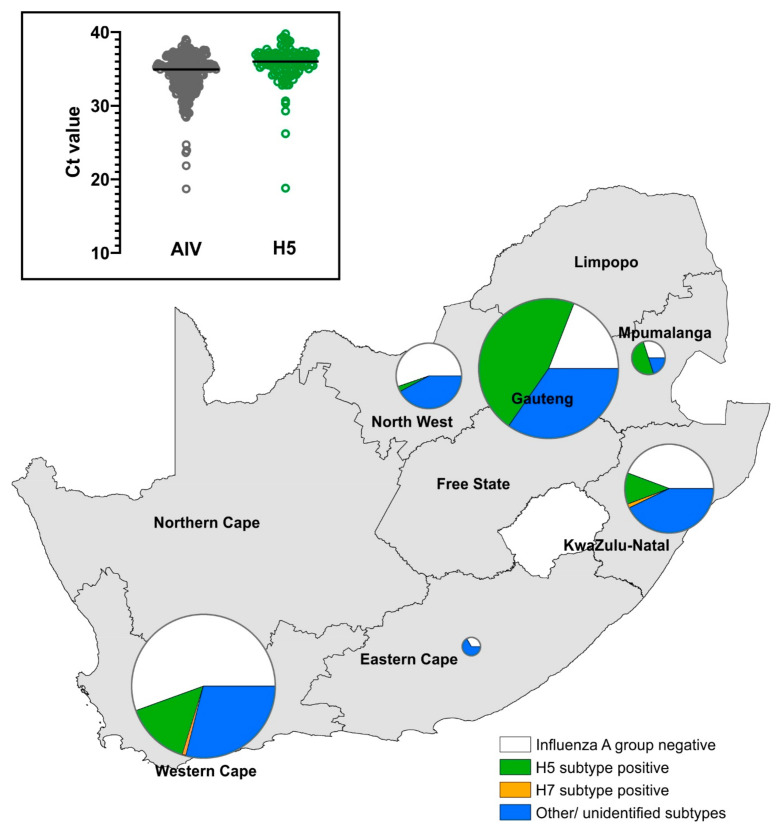
Results of environmental wild bird surveillance in 2023. Pie chart sizes are proportional to the numbers of samples tested. Inset: comparison of the cycle threshold (Ct) values for the real-time rRT-PCR AIV screening assay vs. the H5 subtyping assay.

**Table 1 viruses-16-00896-t001:** Avian influenza viruses (AIV) detected by real-time RT-PCR in wild duck environmental fecal swab pools in South Africa in 2023.

Province	Number of Pools Tested	AIV Positive ^1^	H5 Subtype Positive ^2^	H7 Subtype Positive ^2^	Other/Unidentified Subtypes ^2^
Eastern Cape	3	2 (66.7%)	0	0	2 (100%)
Free State	0	0	0	0	0
Gauteng	173	140 (80.9%)	80 (57.1%)	0	60 (42.9%) ^3^
KwaZulu-Natal	70	39 (55.7%)	8 (20.5%)	1 (2.6%)	30 (76.9%)
Limpopo	0	0	0	0	0
Mpumalanga	10	7 (70%)	5 (71.4%)	0	2 (28.6%)
Northern Cape	0	0	0	0	0
North West	38	17 (44.7%)	1 (5.9%)	0	16 (94.1%)
Western Cape	183	82 (44.8%)	27 (32.9%)	1 (1.2%)	54 (65.9%) ^4^
Total	477	287 (60.1%)	121 (42.2%)	2 (0.7%)	164 (57.1%)

^1^ (percentage of pools tested) ^2^ (percentage of AIV positive pools) ^3^ One pool tested positive for the presence of H11-specific viral RNA. The sample was collected in Muldersdrift on 25 April 2023. ^4^ One pool tested positive for the presence of H9-specific viral RNA. The sample was collected in Oudtshoorn on 3 October 2023.

## Data Availability

All data are presented within the manuscript or as Appendix A. Sequences generated in this study are deposited in the GISAID EpiFlu database under accession numbers: EPI_ISL_18799506 to EPI_ISL18809555. (https://gisaid.org/, accessed 8 May 2024).

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
