# Peer review of "Outbreaks of H5N1 High Pathogenicity Avian Influenza in South Africa in 2023 Were Caused by Two Distinct Sub-Genotypes of Clade 2.3.4.4b Viruses"

_viruses, 2024, doi:10.3390/v16060896_

Round 1

Reviewer 1 Report

Comments and Suggestions for Authors

My views on the article titled "Outbreaks of H5N1 high pathogenicity avian influenza in South Africa in 2023 were caused by two distinct sub-genotypes of clade 2.3.4.4b viruses" are given below:

1. The author should standardize the capitalization of “2.3.4.4b” and “2.3.4.4B” in the main text.

2. Lines 182-184: The viruses described by authors are not clearly indicated in the phylogenic tree. It is recommended that the authors clearly mark the positions of these strains in the phylogenic tree.

3. Line 188: In Figures S1, all strains are depicted in the same color. Could the authors differentiate between the strains used in this study and the reference strains by using different colors?

4. For Figure S3, it is better to display the phylogenetic tree of different segments of all viruses in this study separately.

Wild birds play an important role in the evolution and spread of avian influenza viruses (AIVs). Viruses of clade 2.3.4.4b have attracted a lot of attention recently, through the epidemiological investigation of avian influenza viruses in wild birds, the authors have identified the distribution of different subtypes (including 2.3.4.4b and other low-path AIVs) of avian influenza viruses in wild birds and analyzed the potential transmission chains of avian influenza viruses that exist between different provinces, as well as the impact of environment factors. This article provides data support for monitoring the prevalence of clade 2.3.4.4b and other subtypes of AIVs in wild birds. I congratulate the authors on their successful work.

Author Response

REVIEWER 1

My views on the article titled "Outbreaks of H5N1 high pathogenicity avian influenza in South Africa in 2023 were caused by two distinct sub-genotypes of clade 2.3.4.4b viruses" are given below:

  1. The author should standardize the capitalization of “2.3.4.4b” and “2.3.4.4B” in the main text.

RESPONSE: Corrected in lines 54, 59 and 616, as requested.

  1. Lines 182-184: The viruses described by authors are not clearly indicated in the phylogenic tree. It is recommended that the authors clearly mark the positions of these strains in the phylogenic tree.

RESPONSE: The viruses sequenced in this study (as mentioned in lines 182-184) are actually quite easy to discern in the phylogenetic tree (Fig. 1), because it’s a time-scaled tree and all the new data was from 1 January 2023 onwards (read from the scale at the bottom). No changes made.

  1. Line 188: In Figures S1, all strains are depicted in the same color. Could the authors differentiate between the strains used in this study and the reference strains by using different colors?

RESPONSE: As above, in the time-scaled tree in Fig. S1, the cut-off for newly-sequenced strains vs. reference strains is the 1 January 2023 date, that can be easily read from the timeline at the bottom.  No changes made.

  1. For Figure S3, it is better to display the phylogenetic tree of different segments of all viruses in this study separately.

RESPONSE: It’s not quite clear why this would be “better”, we think it doesn’t make much of a difference since each tree is already referenced as Fig S3(a), (b), (c) etc., and it was uploaded this way for convenience. If the Editor also feels very strongly about this, we will of course make the change. Please advise?

Wild birds play an important role in the evolution and spread of avian influenza viruses (AIVs). Viruses of clade 2.3.4.4b have attracted a lot of attention recently, through the epidemiological investigation of avian influenza viruses in wild birds, the authors have identified the distribution of different subtypes (including 2.3.4.4b and other low-path AIVs) of avian influenza viruses in wild birds and analyzed the potential transmission chains of avian influenza viruses that exist between different provinces, as well as the impact of environment factors. This article provides data support for monitoring the prevalence of clade 2.3.4.4b and other subtypes of AIVs in wild birds. I congratulate the authors on their successful work.

RESPONSE: Thank you.

Reviewer 2 Report

Comments and Suggestions for Authors

The manuscript describes the spread of H5N1 viruses in South Africa from 2021 to 2023.   It is shown that of the 5 clades circulating in 2021, only two have survived to date, and one of them (SA13) contains coastal seabird viruses, while the other (SA15) circulates mainly among poultry. A lot of information interesting and useful for specialists is provided in Supplementary Materials:

The article is well written  and can be accepted in present form.

Author Response

REVIEWER 2

The manuscript describes the spread of H5N1 viruses in South Africa from 2021 to 2023.   It is shown that of the 5 clades circulating in 2021, only two have survived to date, and one of them (SA13) contains coastal seabird viruses, while the other (SA15) circulates mainly among poultry. A lot of information interesting and useful for specialists is provided in Supplementary Materials:

The article is well written and can be accepted in present form.

RESPONSE: Thank you, no further changes made.